# Relation of the ‘Atrial Fibrillation Better Care (ABC) Pathway’ to the Quality of Anticoagulation in Atrial Fibrillation Patients Taking Vitamin K Antagonists

**DOI:** 10.3390/jpm12030487

**Published:** 2022-03-17

**Authors:** Vanessa Roldán, Lorena Martínez-Montesinos, Raquel López-Gálvez, Lucía García-Tomás, Gregory Y. H. Lip, José Miguel Rivera-Caravaca, Francisco Marín

**Affiliations:** 1Department of Hematology and Clinical Oncology, Hospital General Universitario Morales Meseguer, University of Murcia, Instituto Murciano de Investigación Biosanitaria (IMIB-Arrixaca), 30008 Murcia, Spain; vroldans@um.es (V.R.); l.martinez.montesinos@gmail.com (L.M.-M.); lucia.gt47@gmail.com (L.G.-T.); 2Department of Cardiology, Hospital Clínico Universitario Virgen de la Arrixaca, University of Murcia, Instituto Murciano de Investigación Biosanitaria (IMIB-Arrixaca), CIBERCV, 30120 Murcia, Spain; raquellgalvez@gmail.com (R.L.-G.); fcomarino@hotmail.com (F.M.); 3Liverpool Centre for Cardiovascular Science, University of Liverpool and Liverpool Heart and Chest Hospital, Liverpool L14 3PE, UK; gregory.lip@liverpool.ac.uk; 4Department of Clinical Medicine, Aalborg University, 9000 Aalborg, Denmark

**Keywords:** atrial fibrillation, vitamin K antagonists, time in therapeutic range, Atrial Fibrillation Better Care (ABC) pathway

## Abstract

The Atrial Fibrillation Better Care (ABC) pathway was proposed for a more integrated atrial fibrillation (AF) care. We investigated if adherence to the ABC pathway was associated to the quality of anticoagulation control in a cohort of AF outpatients starting vitamin K antagonists (VKAs) between July 2016 and June 2018. Patients were considered adherent to the ABC pathway if they met all of its components. The time in therapeutic range (TTR) was estimated at one year. In total, 1045 patients (51.6% female; median age 77 years; 63% ABC pathway adherent) were included. At one year, 474 (51.6%) of 919 patients with international normalized ratio (INR) data for TTR estimation had a TTR < 65%. Among ABC pathway non-adherent patients, a greater proportion had TRT < 65% (56.4% vs. 43.6%, *p* = 0.025), and TTR < 70% (64.9% vs. 35.1%, *p* = 0.033), with lower mean TTR in non-adherent patients (59.4 ± 22.3% vs. 63.9 ± 21.1%; *p* = 0.004). Logistic regression models demonstrated that the ABC pathway adherence in its continuous (aOR: 0.75, 95% CI 0.59–0.96) and categorical (aOR: 0.75, 95% CI 0.57–0.98) forms was independently associated with TTR ≥ 65%. In this ‘real-world’ cohort of AF patients starting VKAs, the ABC pathway adherent patients had better TTR, and more ABC criteria fulfilled increased the probability of achieving good TTR.

## 1. Introduction

Atrial fibrillation (AF) is the most common arrhythmia, with a prevalence of ~2% in the overall population and up to 15% in the elderly aged ≥80 years old [1]. AF is associated with high morbidity and mortality mainly due to its increased risk of stroke and thromboembolism [2], and oral anticoagulation (OAC, either with vitamin K antagonists [VKAs] or non-vitamin K antagonists [NOACs]) is effective in reducing these risks [3,4]. Despite OAC therapy, cardiovascular complications such as acute coronary syndrome and cardiovascular death are frequent, due to the coexistence of others cardiovascular risk factors such as high blood pressure or diabetes mellitus.

Thus, a more holistic and integrated care approach to managing AF has been proposed, not only focused on stroke prevention, but also efforts to reduce cardiovascular risk factors/comorbidities, including broader approaches such as nurse-led interventions, education and lifestyle modifications (e.g., obesity management, physical exercise and healthy diet) [5,6]. This is proposed in the Atrial Fibrillation Better Care (ABC) pathway, where ‘A’ refers to ‘Avoid stroke’; ‘B’ refers to ‘Better symptom management’ and ‘C’ refers to ‘Cardiovascular and comorbidity risk reduction’ [7]. Recently, the 2020 European Society of Cardiology guidelines on the management of AF, as well as other international guidelines, have recognized this need for a more integrated care, by including for the first time the ABC pathway as a simplified and concise approach that integrates the care of AF patients across various levels of healthcare professionals and between specialties [8,9,10].

Despite the increasing use of NOACs, VKAs are still the most commonly used anticoagulants for stroke prevention in AF in several countries. However, the efficacy and tolerability of VKAs depends on the quality of anticoagulant control, as reflected by the mean time in therapeutic range (TTR) of international normalized ratio (INR) 2.0 to 3.0 [11]. Indeed, a high TTR translates into a lower risk of adverse events [12,13,14]. However, there is no evidence to date whether a more holistic approach to management of AF as reflected by the ABC pathway is associated with better quality of anticoagulation control in VKA users.

In the present study, we investigated the relationship between the ABC pathway and the quality of anticoagulation control in a contemporary cohort of real-world AF patients starting VKA therapy.

## 2. Materials and Methods

Detailed methods of the present study have been previously published [15]. Briefly, this was a prospective observational cohort study including outpatients newly diagnosed with AF and OAC-naïve attending an anticoagulation clinic of a tertiary hospital (Murcia, Spain), from 1 July 2016 to 30 June 2018. The inclusion criteria were as follows: adult AF patients (i.e., ≥18 years old) with documented evidence of AF on ECG and not previously taking OAC for another reason, starting VKAs for the first time. Patients with prosthetic heart valves and severe (mainly rheumatic) valvular AF were excluded. No other exclusion criteria were established.

At baseline, a complete medical history was recorded, including socio-demographic and anthropometric data, comorbidities, concomitant therapies and results of the most recent lab test. Stroke risk (CHA_2_DS_2_-VASc) and bleeding risk (HAS-BLED) were estimated. We also calculated the SAMe-TT_2_R_2_ score [Sex, Age (<60 years); Medical history (at least 2 of the following: hypertension, diabetes, coronary artery disease/myocardial infarction, peripheral arterial disease, congestive heart failure, previous stroke, pulmonary disease, hepatic or renal disease); Treatment (interacting drugs, e.g., amiodarone for rhythm control): all 1 point; the current Tobacco use (2 points); and Race (non-Caucasian; 2 points)] as a measure of whether the patient was likely to have good anticoagulation control on VKA [16].

The study protocol was approved by the Ethics Committee from the University Hospital Morales Meseguer (reference: EST: 20/16), and was carried out in accordance with the ethical standards established in the 1964 Declaration of Helsinki and its subsequent amendments. Informed consent was required for participation in this study.

### 2.1. ABC (Atrial Fibrillation Better Care) Pathway Assessment

The ABC pathway was evaluated according to its original definition, as follows:

*‘A’ Criterion:* At baseline, a patient would qualify for this criterion if properly prescribed and treated with an OAC. As all patients were included in the context of starting VKA therapy and no previous data about the TTR were available, the ‘A’ criterion was considered fulfilled if VKA was correctly prescribed according to thromboembolic risk (i.e., CHA_2_DS_2_-VASc ≥ 1 in males or CHA_2_DS_2_-VASc ≥ 2 in females).

*‘B’ Criterion:* Defined as the presence of symptoms, classified by the European Heart Rhythm Association (EHRA) symptom scale. Any patient with an EHRA score of I (no symptoms) or II (mild symptoms not affecting daily life) qualified for this criterion. Data on symptoms were collected at baseline.

*‘C’ Criterion:* Defined as the optimal management/medical treatment of the main cardiovascular comorbidities: hypertension, coronary artery disease, peripheral artery disease, heart failure, stroke/transient ischaemic attack (TIA), and diabetes mellitus. Optimal medical treatment was defined as follows: (i) for hypertension, this was considered controlled if blood pressure <160/90 mmHg was recorded at baseline and treated with appropriate antihypertensive drugs; (ii) for coronary artery disease, treatment with angiotensin-converting enzyme (ACE) inhibitors, beta-blockers, and statins; (iii) for peripheral artery disease or previous stroke/TIA, treatment with statins; (iv) for heart failure, treatment with ACE inhibitors/angiotensin receptor blockers and beta-blockers; and (v) for diabetes mellitus, treatment with insulin or oral antidiabetics. To be included as adherent to ‘C’ criterion, all main risk factors should have been controlled and/or treated with appropriate drugs.

A patient was considered as fully ABC pathway adherent (‘ABC adherent care’) if all the three criteria were fulfilled.

### 2.2. Follow-Up and Study Outcomes

Follow-up was performed according to the standard of care at each routine visit to the outpatient anticoagulation clinic or visits for the anticoagulation control. Medical records and telephone calls were used to obtain the information needed and vital status, if the patient never attends to these visits. No specific interventions and no specific visits were performed for study purposes. During one-year of follow-up, all INR measurements were recorded. The therapeutic range was established at INR 2.0–3.0 according to national and international recommendations.

For the present study, the primary endpoint was the quality of anticoagulation with VKA by using the TTR calculated by the linear interpolation method of Rosendaal at one-year after entry [17]. The linear interpolation proposed by Rosendaal is based on the assumption that moving from one INR to a different one in two determinations separated by a certain number of days occur in a linear way, crossing the difference between the two INR values during those days. Thus, it estimates that the difference between two different INR determinations belonging to different days was produced by increasing or decreasing the INR each day [17]. The secondary endpoint was the quality of anticoagulation with VKA by using the proportion of INRs in range (PINRR, the so-called direct method). This simple method estimates the quality of anticoagulation by taking into account how many INRs are within the therapeutic range (i.e., INR 2.0–3.0) over the total INRs measured [18]. The TTR and PINRR were calculated, excluding the first month of anticoagulation, and we used two cut-off points for the definition of not well controlled VKA therapy (<65% and <70%).

### 2.3. Statistical Analyses

Continuous variables were expressed as mean ± standard deviation (SD) or median and interquartile range (IQR) as appropriate, whilst categorical variables were expressed as absolute frequencies and percentages. The Pearson chi-squared test was used to compare proportions, and differences between continuous and categorical variables were assessed using the Mann–Whitney U test or the Student *t* test, as appropriate. The correlation between the ABC pathway and TTR was tested using the Pearson’s r.

Multivariate logistic regression analyses were performed to determine the association between the ABC pathway and the primary/secondary endpoints. A univariate significance level of 0.05 was required to allow a variable into the multivariate model (SLENTRY = 0.05) and a multivariate significance level of 0.05 was required for a variable to stay in the model (SLSTAY = 0.05). Results were reported as adjusted odds ratios (aOR) with a 95% confidence interval (CI).

A *p*-value < 0.05 was accepted as statistically significant. Statistical analyses were performed using SPSS v. 25.0 (SPSS, Inc., Chicago, IL, USA), and MedCalc v. 16.4.3 (MedCalc Software bvba, Ostend, Belgium) for Windows.

## 3. Results

Overall, 1254 patients with AF were initially screened and 1064 were included. Of these, 14 patients were lost to follow-up and 5 patients had no all data for the ABC pathway estimation, giving a final study cohort of 1045 patients (Figure 1) (51.6% female; median age 77, IQR 70–83 years) with a median CHA_2_DS_2_-VASc of 4 (IQR 3–5) and HAS-BLED of 2 (IQR 2–3). Baseline data are summarized in Table 1.

With regards to the ABC pathway, 1017 (97.3%) of patients fulfilled the “A” criterion at baseline; 890 (85.2%) fulfilled the “B” criterion; and 809 (77.4%) fulfilled the “C” criterion. Overall, 32 (3.1%) were adherent to one criterion, 355 (34.0%) were adherent to two criteria, and 658 (63.0%) were adherent to all three criteria. Thus, 658 (63%) were categorized as adherent to the ABC pathway at baseline, whereas 387 (37%) were considered not adherent. 

However, enough INR data for TTR estimation was available in 919 patients at one-year of follow-up (Figure 1). The mean TTR of these patients was 62.3% ± 21.7%; 474 (51.6%) of them did not achieve a TTR over 65% (mean TTR 45.9% ± 16.2%) and 555 (60.4%) did not achieve a TTR over 70% (mean TTR 49.0% ± 16.8%). Among those non-adherent to the ABC pathway at baseline, a higher proportion presented a TTR < 65% than TTR ≥ 65% (56.4% vs. 43.6%, *p* = 0.023), as well as a higher proportion had TTR < 70% compared to TTR ≥ 70% (64.9% vs. 35.1%, *p* = 0.031) at one year. Of note, the mean TTR was lower in those non-adherent to the ABC pathway compared to ABC-adherent patients (59.4% ± 22.3% vs. 63.9% ± 21.1%; *p* = 0.002).

### 3.1. ABC Pathway Adherence and Quality of Anticoagulation Control

The ABC pathway as a continuous parameter and the TTR were significantly correlated (*p* < 0.001) (Figure 2). A logistic regression model showed that the ABC pathway in its continuous form was associated with the quality of anticoagulation control, even after adjusting for the SAMe-TT_2_R_2_ score. A greater number of ABC pathway criteria fulfilled was independently associated with a TTR ≥ 65% (adjusted OR 0.75, 95% CI 0.59–0.96, *p* = 0.020), and with a TTR ≥ 70% (aOR 0.73, 95% CI 0.57–0.94, *p* = 0.015).

Patients adherent to the three ABC pathway criteria had a significantly higher probability of achieving TTR ≥ 65% (aOR 0.41, 95% CI 0.18–0.95, *p* = 0.038) when compared to patients adherent to one or two criteria only. Fulfilling the “C” criterion alone was also significantly related to TTR ≥ 65% (aOR 0.67, 95% CI 0.49–0.91, *p* = 0.010). Importantly, the results were consistent when using the TTR ≥ 70% as the cut-off (Table 2).

In its categorical form (i.e., adherent vs. non-adherent), the ABC pathway was significantly associated with the quality of anticoagulation control in the model adjusted for SAMe-TT_2_R_2_. Patients categorized as adherent to the ABC pathway had a higher probability of TTR ≥ 65% (aOR 0.75, 95% CI 0.57–0.98, *p* = 0.039), although a non-significant trend was seen when the cut-off point for TTR was 70% (aOR 0.77, 95% CI 0.58–1.02, *p* = 0.065) (Figure 3).

### 3.2. ABC Pathway Compliance and PINRR

With regard to the secondary outcome, after adjusting for the SAMe-TT_2_R_2_ score, the ABC pathway was independently associated with a PINRR ≥ 65%, with an aOR of 0.76 (95% CI 0.59–0.98, *p* = 0.044) for the continuous form and an aOR of 0.72 (95% CI 0.54–0.97, *p* = 0.029) for the categorical form (adherent vs. non-adherent). This was not statistically significant for PINRR ≥ 70% (aOR 0.76, 95% CI 0.56–1.02, *p* = 0.067 and aOR 0.75, 95% CI 0.54–1.04, *p* = 0.087; respectively).

## 4. Discussion

In this study, we demonstrated that an integrated management according to the ‘ABC pathway’ could lead to better anticoagulation therapy with VKAs in terms of TTR. This is central since the efficacy and safety of VKA depends on the quality of anticoagulant control, as reflected by the average TTR of INRs 2.0 to 3.0. Various studies have shown how a high TTR translates into a lower risk of stroke and bleeding while on OAC [12,13,19]. The maintenance dose of VKA is influenced by many different factors, including race, dietary vitamin K intake, comorbidities (e.g., liver disease and acute illness), or whether the patient may be taking interacting drugs [20]. The average individual TTR range generally increases over time, but even in very well and experienced patients, the TTR can decrease during follow-up [3]. In this context, drug adherence is important in TTR maintenance; indeed, non-adherence and low anticoagulation control levels are associated [21,22]. Adherence is also linked to a better knowledge of the patient about its disease and the treatment. In patients taking warfarin in an anticoagulation clinic who completed a questionnaire survey about knowledge, satisfaction and concerns regarding warfarin treatment, those with better knowledge and higher satisfaction were those with higher warfarin adherence and better INR control [23,24,25].

The SAMe-TT_2_R_2_ score was proposed to identify those AF patients that would be less likely to do well with good anticoagulation control on VKAs [16]. As the SAMe-TT_2_R_2_ score includes several comorbidities/cardiovascular risk factors, it not only predicts poor anticoagulation control but also adverse events [26]. Indeed, the SAMe-TT_2_R_2_ score sums up the influence of comorbidities and adjunctive treatment on the response to VKA treatment. The ABC pathway goes beyond this, since it demonstrates that correctly managed AF patients and the ABC-adherent might have higher TTR, beyond the presence of comorbidities alone. In this setting, we demonstrated in the present study that the ABC pathway is associated with well-managed VKA treatment, independently of the SAMe-TT_2_R_2_ score.

Despite the limitations of VKAs (narrow therapeutic range, multiple food and drugs interactions), investing in education and counselling would improve the quality of anticoagulation control amongst VKA-anticoagulated patients [27]. One pilot study showed that even a brief educational intervention can help to improve the knowledge about anticoagulation therapy for AF, focused on patient’s knowledge of the target INR range and factors that may affect INR levels [28].

Since its first publication, several studies have demonstrated that ABC pathway-adherent-patients have better outcomes that those who are not adherent [29,30,31]. In a systematic review including eight studies and more than 285,000 AF patients, there was a pooled prevalence of 21% who were ABC-adherent patients [32]. Importantly, adherence to the ABC pathway was associated with a reduction in the risk of major adverse outcomes [32]. However, most of the evidence about the ABC pathway today derives from NOAC-treated AF cohorts. Nevertheless, a low TTR is associated with worse outcomes [33], including higher risk of mortality [13]. Thus, TTR and clinical outcomes are intimately related. In this context, our study addresses the fact that patients who fulfilled the ABC pathway showed a higher TTR value perhaps because of better clinical management. AF patients are considered as a clinical complex group with multiple comorbidities, polypharmacy and multiple cardiovascular events, so a holistic approach would result in better management and a reduction not only for worse clinical outcomes but also for higher quality of anticoagulation control with VKA therapy. Despite NOACs have changed the landscape of OAC in AF patients, there are still regional differences in the prescription of these drugs (for example, in Spain, whereby NOACs are only reimbursed if AF patients fulfill very specific conditions). There are also patients for whom NOACs are contraindicated, for example, AF patients with a mechanical heart valve, who need good anticoagulation control with VKAs. The PLECTRUM study, which includes 2111 patients with mechanical heart valves, showed that hypertension, diabetes and heart failure were independently associated with a low TTR, providing clinical evidence that comorbidities directly affect anticoagulant management and better INR control [34].

In summary, adherence with ABC pathway management is not only associated with a lower number of cardiovascular events but also results in better anticoagulation control, which in turn is associated with a better prognosis. This strategy is useful for both types of OACs, as NOACs require a close follow-up to ensure good adherence, while VKAs need a high TTR for optimal clinical benefit.

### Limitations

Our study has some limitations. The main limitation of the study lies in its observational nature, with a Caucasian-based population and single centre design, performed in a unique anticoagulation clinic. It has been previously demonstrated that anticoagulation clinics have better results in terms of the TTR achieved [35], so data from other anticoagulation management models should be explored. We included patients who were VKA-naïve, which has two drawbacks: first, the start of anticoagulation with VKA is a complex period, with a greater number of adverse events; and second, not all patients reach the therapeutic range at the same time.

## 5. Conclusions

In this ‘real world’ prospective cohort of AF patients starting VKA therapy, management adherent to the ABC pathway was related to the quality of anticoagulation control at one year. Adherent patients to the ABC pathway had better TTR, and more ABC criteria fulfilled were associated with higher probability of achieving good TTR.

## Figures and Tables

**Figure 1 jpm-12-00487-f001:**
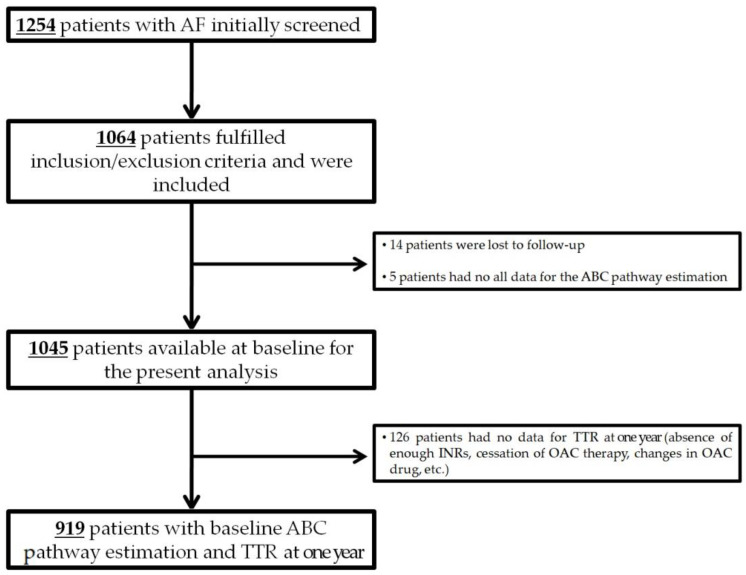
Flow-chart of the study. AF = atrial fibrillation; INR = international normalized ratio; OAC = oral anticoagulation; TTR = time in therapeutic range.

**Figure 2 jpm-12-00487-f002:**
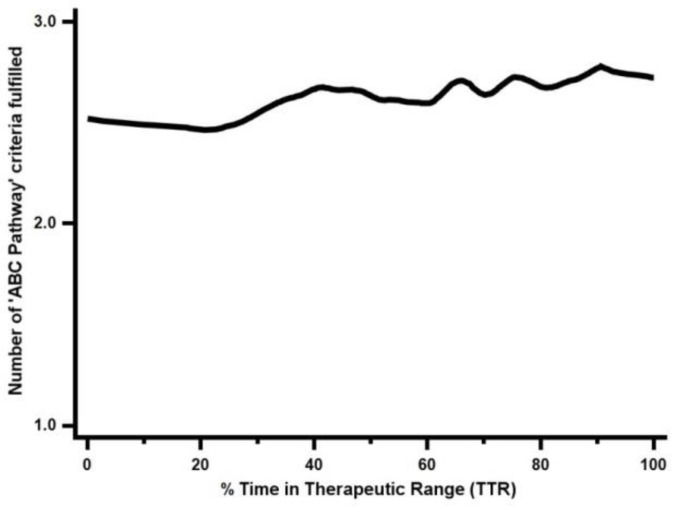
Scatter diagram showing the graphical correlation of the ABC pathway and TTR.

**Figure 3 jpm-12-00487-f003:**
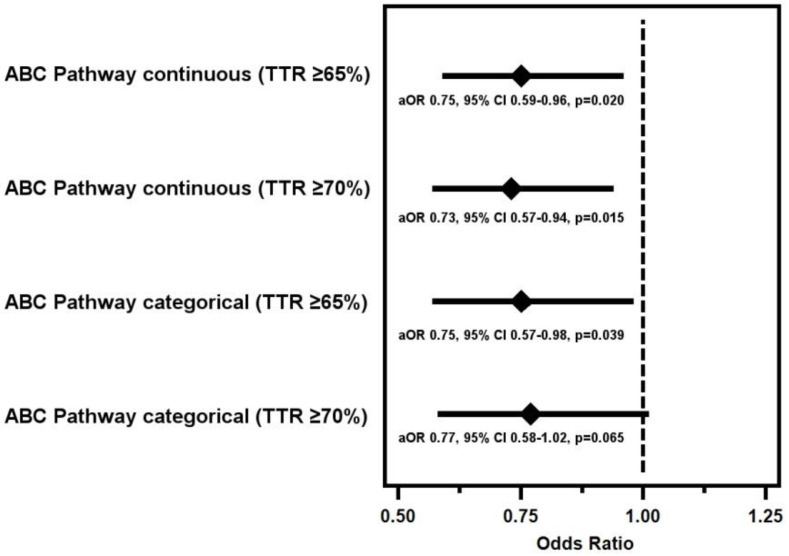
Probability of poor TTR in relation to the ABC pathway.

**Table 1 jpm-12-00487-t001:** Baseline clinical characteristics.

	N = 1045
**Demographic**	
Male sex, n (%)	506 (48.4)
Age (years), median (IQR)	77 (70–83)
BMI (kg/m^2^), median (IQR)	30.0 (26.8–33.3)
**Comorbidities, n (%)**	
Hypertension	874 (83.6)
Diabetes mellitus	393 (37.6)
Heart failure	261 (25.0)
History of stroke/TIA/thromboembolism	162 (15.5)
Renal impairment	197 (18.9)
Coronary artery disease	190 (18.2)
Peripheral artery disease	66 (6.3)
Hypercholesterolemia	608 (58.2)
Current smoking habit	157 (15.0)
Current alcohol consumption	71 (6.8)
History of previous bleeding	173 (16.6)
COPD/OSAH	230 (22.0)
Hepatic disease	68 (6.5)
Concomitant malignant disease	150 (14.4)
**Concomitant treatment, n (%)**	
Antiarrhythmics	214 (20.5)
Calcium antagonist	317 (30.3)
Beta-blockers	723 (69.2)
Statins	555 (53.1)
Diuretics	571 (54.6)
Antiplatelet therapy	256 (24.5)
ACE inhibitors	255 (24.4)
Angiotensin II receptor blockers	456 (43.6)
Oral antidiabetics/insulin	279 (26.7)
CHA_2_DS_2_-VASc, median (IQR)	4 (3–5)
HAS-BLED, median (IQR)	2 (2–3)
SAMe-TT_2_R_2_, median (IQR)	1 (1–2)

ACE inhibitors = angiotensin-converting enzyme inhibitors; COPD/OSAH = chronic obstructive pulmonary disease/obstructive sleep apnoea/hypopnoea; BMI = body mass index; IQR = interquartile range; TIA = transient ischemic attack.

**Table 2 jpm-12-00487-t002:** Association of ABC components with quality of anticoagulation therapy by different thresholds.

	TTR < 65%		TTR < 70%	
	aOR (95% CI)	*p*-Value	aOR (95% CI)	*p*-Value
ABC pathway (1 criterion)	Ref.		Ref.	
ABC pathway (2 criteria)	0.51 (0.22–1.20)	0.125	0.30 (0.10–0.88)	0.028
ABC pathway (3 criteria)	0.41 (0.18–0.95)	0.038	0.25 (0.08–0.72)	0.010
ABC pathway (A criterion)	0.56 (0.24–1.33)	0.191	0.42 (0.16–1.15)	0.092
ABC pathway (B criterion)	1.02 (0.70–1.48)	0.923	0.99 (0.68–1.46)	0.974
ABC pathway (C criterion)	0.67 (0.49–0.91)	0.010	0.66 (0.48–0.92)	0.013

aOR = adjusted odds ratio; CI = confidence interval; TTR = time in therapeutic range.

## Data Availability

Data cannot be shared for ethical/privacy reasons.

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
