# Peer review of "Relation of the ‘Atrial Fibrillation Better Care (ABC) Pathway’ to the Quality of Anticoagulation in Atrial Fibrillation Patients Taking Vitamin K Antagonists"

_jpm, 2022, doi:10.3390/jpm12030487_

Round 1

Reviewer 1 Report

The article discusses a very important issue, which is the problem of adherence in the group of patients with atrial fibrillation. Due to the frequent use of VKA in this group of patients and the need to monitor anticoagulant treatment, it is important to introduce models of comprehensive care for these patients. An example of such care is the ABC model presented, which indicates the need for interdisciplinary patient care. This approach was presented in the reviewed article, which is a significant value of the work.

The presented results come from the study, the protocol of which was previously published in the peer-reviewed journal [footnote 15] - the presented methodology of the study does not raise any objections.

The applied statistical tests are adequate to the analyzed data.

In terms of the presentation of the results - In my opinion, it is unnecessary to combine two figures into one (Figure 1 A and 1B). It would be more appropriate to divide them into two separate ones. Moreover, it is worth showing the value of the p value in the figures.

Moreover, in my opinion, it is unnecessary to repeat the results in lines 180, 182 and 183, since the same data is presented in the table under line 185. It would be more adequate to indicate a specific range of ABC and TTR.

The discussion is comprehensive and well structured. However, I am skeptical about the first paragraph of the discussion, which in my opinion is redundant and is more a description of the conclusions.

Author Response

The article discusses a very important issue, which is the problem of adherence in the group of patients with atrial fibrillation. Due to the frequent use of VKA in this group of patients and the need to monitor anticoagulant treatment, it is important to introduce models of comprehensive care for these patients. An example of such care is the ABC model presented, which indicates the need for interdisciplinary patient care. This approach was presented in the reviewed article, which is a significant value of the work.

>>> Thank you very much for your overall positive opinion about our study.

The presented results come from the study, the protocol of which was previously published in the peer-reviewed journal [footnote 15] - the presented methodology of the study does not raise any objections.

>>> Thanks for this comment.

The applied statistical tests are adequate to the analyzed data.

>>> Thank you.

In terms of the presentation of the results - In my opinion, it is unnecessary to combine two figures into one (Figure 1 A and 1B). It would be more appropriate to divide them into two separate ones. Moreover, it is worth showing the value of the p value in the figures.

>>> We agree with you regarding this suggestion. Figures 1A and 1B have now been divided into two different figures (now Figure 2 and Figure 3), and p-values have been added to Figure 3. 

Moreover, in my opinion, it is unnecessary to repeat the results in lines 180, 182 and 183, since the same data is presented in the table under line 185. It would be more adequate to indicate a specific range of ABC and TTR.

>>> We apologize for this issue. We have modified the text according to your suggestion.

The discussion is comprehensive and well structured. However, I am skeptical about the first paragraph of the discussion, which in my opinion is redundant and is more a description of the conclusions.

>>> Thanks for this comment. We have modified that paragraph.

Reviewer 2 Report

The authors investigated if adherence to the ABC pathway related to the quality of anticoagulation control in a cohort of AF outpatients starting vitamin K antagonists (VKAs) between July 2016 and June 2018 including 1045 patients. The time in therapeutic range (TTR) was estimated at 1-year. 474 patients had a TTR <65%, and 555 had a TTR <70%. Logistic regression models demonstrated that the ABC pathway adherence in its continuous (aOR: 0.75, 95% CI 0.59-0.96) and categorical (aOR: 0.75, 95% CI 0.57-0.98) forms were independently associated with TTR ≥65%. In this ‘real world’ cohort of AF patients starting VKAs, adherent patients to the ABC pathway had better TTR and more ABC criteria fulfilled. They were associated with higher probability of achieving good TTR.

Dear authors,

you present a very interesting study. However, there are some drawbacks to consider that we believe hinder an immediate publication.

The heading and especially the abstract of a manuscript should present the content to the reader in an understandable way and at the same time attract the reader to study the whole text.

Headline: The headline is somewhat difficult to understand. Please improve.

Abstract: The abstract needs an urgent revision. E.g., in my opinion, the figures and calculations in lines 22-24 are inconsistent and also difficult to understand. Line 22: "...1045 patients 22 (51.6% female; median age 77 years; 63% ABC pathway adherent)". The sentence seems to be incomplete and needs improvement. Line 23-24: 1045 patients 22 (51.6% female; median age 77 years; 63% ABC pathway adherent): 51,5% might be incorrect. If not, please explain.

Methods: acceptabel

Results: In need of improvement. In particular, the calculations of lines 161 to 163 were unclear. The total population is 918.6 points when 474 points were 51.6% of it.  Please explain the 918.6 pts. All in all, a flow chart would be helpful, starting with the 1045 points.

Figure 1 also needs to be improved. In addition to better labeling, the black border should be removed.

The graphical representation of the tables should be improved, at least in my version it is not consistent and poor.

Overall, it is well worked out that an improvement of the measured values can be achieved by the ABC pathway. What is missing, however, are the clinical results, such as stroke, myocardial infarction, bleeding, death etc. This is precisely the key point of better measuring. In the controls after one year you asked for adverse events. Therefore these data should be included in the manuscript.

Measuring alone is not enough. For example, many studies on INR self-monitoring have shown that although the measured values were better, there was no difference in adverse events between the groups.

Discussion: acceptable.

Literature: acceptable

Author Response

Reviewer: 2

The authors investigated if adherence to the ABC pathway related to the quality of anticoagulation control in a cohort of AF outpatients starting vitamin K antagonists (VKAs) between July 2016 and June 2018 including 1045 patients. The time in therapeutic range (TTR) was estimated at 1-year. 474 patients had a TTR <65%, and 555 had a TTR <70%. Logistic regression models demonstrated that the ABC pathway adherence in its continuous (aOR: 0.75, 95% CI 0.59-0.96) and categorical (aOR: 0.75, 95% CI 0.57-0.98) forms were independently associated with TTR ≥65%. In this ‘real world’ cohort of AF patients starting VKAs, adherent patients to the ABC pathway had better TTR and more ABC criteria fulfilled. They were associated with higher probability of achieving good TTR.

Dear authors,

You present a very interesting study. However, there are some drawbacks to consider that we believe hinder an immediate publication.

The heading and especially the abstract of a manuscript should present the content to the reader in an understandable way and at the same time attract the reader to study the whole text.

Headline: The headline is somewhat difficult to understand. Please improve.

>>> Thank you for this interesting suggestion. We have modified the title and we hope it is now clearer.

Abstract: The abstract needs an urgent revision. E.g., in my opinion, the figures and calculations in lines 22-24 are inconsistent and also difficult to understand. Line 22: "...1045 patients 22 (51.6% female; median age 77 years; 63% ABC pathway adherent)". The sentence seems to be incomplete and needs improvement. Line 23-24: 1045 patients 22 (51.6% female; median age 77 years; 63% ABC pathway adherent): 51,5% might be incorrect. If not, please explain.

>>> Thank you very much for your attention. We really apologize for the issues found by the reviewer. We have revised the abstract and corrected when appropriate.

Methods: acceptable

>>> Thank you.

Results: In need of improvement. In particular, the calculations of lines 161 to 163 were unclear. The total population is 918.6 points when 474 points were 51.6% of it. Please explain the 918.6 pts. All in all, a flow chart would be helpful, starting with the 1045 points.

>>> We really appreciated this suggestion. We have clarified some issues in the Results section and added a flow chart of the study (now Figure 1).

Figure 1 also needs to be improved. In addition to better labeling, the black border should be removed.

>>> We agree with the reviewer regarding this suggestion and therefore we have clarified the title of the figure. We have also removed the black border and following the suggestion of the reviewer 1, we have separated this into two different figures (now Figure 2 and Figure 3).

The graphical representation of the tables should be improved, at least in my version it is not consistent and poor.

>>> We apologize for this issue. The tables have been formatted by the editorial office according to the journal’s guidelines.

Overall, it is well worked out that an improvement of the measured values can be achieved by the ABC pathway. What is missing, however, are the clinical results, such as stroke, myocardial infarction, bleeding, death etc. This is precisely the key point of better measuring. In the controls after one year you asked for adverse events. Therefore these data should be included in the manuscript.

Measuring alone is not enough. For example, many studies on INR self-monitoring have shown that although the measured values were better, there was no difference in adverse events between the groups.

>>> Thank you for this interesting comment. We agree regarding the fact that the association of the ABC Pathway and clinical outcomes is relevant and should be investigated. However, that was not the aim of the present study. Whereas previous studies have already demonstrated that adherence to the ABC Pathway reduces the risk of worse clinical outcomes, our primary outcome was the quality of anticoagulation with VKA by using the TTR, which is a completely novel approach. Thus, we were not looking here for potential associations between the ABC Pathway and clinical outcomes but if an integrated management according to the ‘ABC Pathway’ leads to better anticoagulation control.

However, we fully agree with you, and indeed the relation of ABC Pathway and clinical outcomes has been already described in another manuscript that is currently under consideration in a different journal. To avoid dual publication, we cannot include such results in the present manuscript, but we provide them here for the reviewer. In summary, after adjusting for several comorbidities, the ABC adherent care favored the reduction of the risk of all-cause mortality (aHR 0.57, 95% CI 0.42-0.78), cardiovascular mortality (aHR 0.54, 95% CI 0.32-0.90), and net clinical outcomes (aHR 0.72, 95% CI 0.56-0.92) (defined as the composite of major bleeding, ischemic stroke/TIA, and all-cause mortality).

Discussion: acceptable.

>>> Thank you.

Literature: acceptable

>>> Thank you.